# Aβ_40_ Improves Cerebrovascular Endothelial Function via NOX4-Dependent Hydrogen Peroxide Release

**DOI:** 10.3390/ijms26146759

**Published:** 2025-07-15

**Authors:** Elizabeth Heller, Lindsey McGurran, Joseph K. Brown, Kathleen Love, Matthew Hobbs, Jeong Sook Kim-Han, Byung Hee Han

**Affiliations:** Department of Pharmacology, Kirksville College of Osteopathic Medicine, A.T. Still University of Health Sciences, Kirksville, MO 63501, USA; sa209255@atsu.edu (E.H.); sa211165@atsu.edu (L.M.); jkeeganbrownkcom@atsu.edu (J.K.B.); sa212464@atsu.edu (M.H.); jeongkimhan@atsu.edu (J.S.K.-H.)

**Keywords:** Alzheimer’s disease, amyloid beta, endothelial cells, NADPH oxidase, reactive oxygen species, hydrogen peroxide, superoxide, nitric oxide

## Abstract

Alzheimer’s disease (AD) is associated with an abnormal accumulation of amyloid β (Aβ) fibrils in the brain parenchyma and cerebrovasculature, which leads to cognitive impairment and cerebrovascular dysfunction. Cerebrovascular endothelial cells play a crucial role in regulating cerebral blood flow, vascular permeability, and neurovascular function. Reactive oxygen species (ROS), particularly those generated by nicotinamide adenine dinucleotide phosphate (NADPH) oxidase 2 (NOX2), contribute to vascular dysfunction and amyloid deposition in the Alzheimer’s disease (AD) brain. However, the role of the NOX4 isoform in AD pathogenesis remains to be examined. In the present study, we found that NOX4 among the NOX isoforms is predominantly expressed in bEnd.3 mouse brain endothelial cells. Treatment with Aβ_40_ significantly enhanced the release of H_2_O_2_ and NO, and increased the endothelial cell viability. To test the involvement of NOX4 in Aβ_40_-induced H_2_O_2_ production, we utilized pharmacological inhibitors of NOX isoforms. Aβ_40_-induced H_2_O_2_ production was attenuated in the presence of the pan-NOX inhibitor, apocynin, or the NOX1/4-selective inhibitors, setanaxib and GKT136901. Since only the NOX4 isoform is expressed in bEnd.3 cells, these results indicate that NOX4 is responsible for the release of H_2_O_2_ stimulated by Aβ_40_. Taken together, the present study demonstrated that Aβ_40_ peptide exerts beneficial effects in bEnd.3 endothelial cells via the NOX4-dependent mechanism.

## 1. Introduction

Alzheimer’s disease (AD) is a neurodegenerative disease that is the leading cause of dementia and globally has risen to the seventh leading cause of death [1,2]. Limited treatment options warrant further research into a potential treatment for AD. One hallmark characteristic of AD includes an abnormal accumulation of amyloid β (Aβ) fibrils in the brain parenchyma and cerebrovasculature, causing neurovascular dysfunction [3,4]. Aβ (major forms: Aβ_40_ and Aβ_42_) is produced as a soluble monomer from the amyloid precursor protein (APP) through multiple enzymatic cleavage events [5]. The impaired clearance of the Aβ peptide contributes to its aggregation and accumulation in the AD brain. Within the vessels in the brain, Aβ_40_ is the Aβ species mostly deposited as a form of cerebral amyloid angiopathy (CAA) [6,7]. While AD and CAA are clinically viewed as separate diseases, the high rate of occurrence of CAA in AD patients (85–95%) suggests that a connection exists between the two [8]. CAA significantly impacts cerebrovascular morphology and function, resulting in weakened vessel walls and dysregulation of cerebral blood flow (CBF) [9].

A growing body of evidence indicates that soluble Aβ monomers cause cerebrovascular dysfunction, while deposition of insoluble Aβ fibrils in the form of CAA further leads to cerebrovascular impairment. For example, soluble Aβ_40_ peptide (and Aβ_42_ to a lesser degree) induces direct vasoconstriction and alters the cerebrovascular reactivity to vasoconstrictors and vasodilators in isolated cerebral arterioles [10,11,12,13]. Similarly, superfusion of Aβ_40_ on the neocortical cortex significantly reduces resting CBF induced by the endothelium-dependent vasodilators, but this peptide does not affect CBF change induced by the endothelium-independent vasodilators [14,15]. Furthermore, we have previously reported that elevated levels of soluble Aβ reduce cerebrovascular reactivity, which can be partially restored by attenuating the production of soluble Aβ [13].

Cerebrovascular endothelial cells (CECs) play a critical role in maintaining homeostasis of the central nervous system. They form a specialized membrane around blood vessels that underlies the blood–brain barrier (BBB) [16,17,18]. CECs are also involved in maintaining the cerebrovascular tone that regulates regional CBF in response to local neural activity. A key mediator synthesized and released by CECs includes the potent vasodilator, nitric oxide (NO), which is mainly produced by endothelial nitric oxide synthases (eNOSs) in the cerebral vessels [19,20]. The bioavailability of NO is important for the maintenance of brain function, and impaired NO bioavailability is associated with various neurodegenerative diseases. Reactive oxygen species (ROS) such as superoxide are known to contribute to a decrease in the NO bioavailability associated with brain pathophysiology, including AD [21].

Recent studies have shown that Aβ increases the production of ROS, which can lower the bioavailability of NO in human and animal models of AD [22,23]. ROS are highly reactive oxygen-containing molecules, including superoxide radical (O_2_^•−^), hydroxyl radical (^•^OH), and hydrogen peroxide (H_2_O_2_). Superoxide radical is produced by redox enzymes involved in the mitochondrial respiratory chain, nicotinamide adenine dinucleotide phosphate (NADPH) oxidases, and xanthine oxidases. This cytotoxic, chemically reactive superoxide can be converted to less cytotoxic hydrogen peroxide radical by superoxide dismutase. Hydrogen peroxide can be converted to a highly reactive hydroxyl radical through Fenton’s reaction, or it can be broken down to water and oxygen by catalases [23,24,25].

NADPH oxidases (NOX) play a central role in the production of ROS under normal physiological as well as pathophysiological conditions [25,26,27,28]. They include NOX1, NOX2, NOX3, NOX4, DUOX1, and DUOX2 isoforms [25]. All NOX isoforms share the conserved structural properties with at least six transmembrane domains, a flavin adenine dinucleotide (FAD), and NADPH-binding cytosolic domains that act as a catalyst for the transfer of two electrons from NADPH through their FAD domain and two haem prosthetic groups to molecular oxygen [25]. NOX1, NOX2, and NOX3 isoforms interact with p22^phox^ transmembrane protein along with the cytosolic subunits, including the activator rac GTPase, generating primarily superoxide radical [29]. NOX4 only requires the p22^phox^ subunit for its catalytic activity, which predominantly produces hydrogen peroxide [29,30,31]. NOX isoforms are ubiquitously expressed in many different tissues throughout the body, including the brain vessels [6,29].

We and others have demonstrated that NOX2 is the major NADPH oxidase isoform that mediates Aβ-induced cerebrovascular dysfunction [6,32]. For example, we found that the *Nox2* gene was highly expressed in cerebral arteries isolated from 12-month-old mice. Its expression was significantly increased in the Tg2576 transgenic mouse model of AD as compared with wild-type littermates [6]. The non-selective NOX inhibitor, apocynin, was not only able to restore cerebrovascular dysfunction but also decreased the Aβ deposition in the blood vessels [6]. In line with this, Park et al. observed that there was a significant decrease in ROS levels, cerebrovascular dysfunction, and behavioral deficits in old Tg2576 mice lacking NOX2 expression as compared with age-matched wild-type mice [33], indicating the critical role of NOX2-driven ROS in AD pathophysiology. However, the role of other NOX isoforms, particularly NOX4, in AD pathophysiology remains to be explored. Recently, selective NOX inhibitors with small molecular weights have been reported [34]. In the present study, we examined the involvement of NOX4 in mediating cerebrovascular actions of Aβ peptide in cultured endothelial cells.

## 2. Results

### 2.1. Aβ_40_ Increases the Release of Hydrogen Peroxide and Cell Viability in bEnd.3 Cells

We utilized the amplex red assay method to measure the levels of hydrogen peroxide. In our cell-free assay condition, the amplex red fluorescent signals increased in the presence of hydrogen peroxide in a concentration-dependent manner, while these fluorescent signals were completely abolished in the presence of catalase which hydrolyzes hydrogen peroxide (Figure 1A). In addition, this assay allowed us to detect as little as 0.2 µM hydrogen peroxide. When endothelial cells were incubated with various concentrations of Aβ_40_ for 3 h, we found that treatment with 1 µM and 2 µM Aβ_40_ significantly increased the release of hydrogen peroxide (Figure 1B). To test the effect of Aβ_40_ peptide on bEnd.3 cell viability, cells were incubated for 24 h with various concentrations of Aβ_40_ in the L-15 culture medium deprived of fetal bovine serum. We found that incubation with Aβ_40_ at concentrations of 2 and 4 μM significantly increased the cell viability as compared with the control group in a concentration-dependent manner (110.1 ± 2.4%, and 110.8 ± 3.4% vs. 100 ± 1.9%, *p* < 0.05) (Figure 1C). When the cell viability assay was performed in the same condition, incubation with 10 and 100 µM hydrogen peroxide significantly increased the endothelial viability whereas treatment with 10 mM hydrogen peroxide led to cell death by almost 100% (Figure 1D).

### 2.2. Aβ_40_ Enhances the NO Bioavailability in bEnd.3 Cells

We utilized the NO-sensitive fluorescent dye, DAF-AM, to determine the effects of Aβ_40_ on NO bioavailability. When endothelial cells were incubated with various concentrations of Aβ_40_ for 3 h, this peptide at concentrations of 2 and 4 µM significantly increased the NO levels in our assay condition (Figure 2A). Next, we incubated bEnd.3 cells with various concentrations of hydrogen peroxide, and the NO production was assessed after 3 h of incubation. There was a significant increase in the NO production when cells were treated with 100 µM hydrogen peroxide (Figure 2B).

### 2.3. Nox4 Is the Nox Isoform That Is Predominantly Expressed in bEnd.3 Cells

Since different NOX isoforms are ubiquitously present in many different types of cells, including cerebrovascular endothelial cells, we wanted to identify Nox isoforms expressed in the bEnd.3 endothelial cells. Our results from the RT-qPCR study reveal that *Nox4* mRNA is highly abundant by approximately 4% of that of *Gapdh* (Figure 3A), whereas the levels of *Nox-1*, *-2*, and *-3* mRNA expression were negligible. When bEnd.3 cells were treated with 2 μM Aβ_40_, the gene expression levels of *Nox* isoforms remained unchanged up to 24 h of the treatment (*p* > 0.05) (Figure 3B).

### 2.4. NOX4 Is Responsible for Increased Hydrogen Peroxide Production in Response to Aβ_40_

To test if NOX4 contributes to the production of hydrogen peroxide induced by Aβ_40_ peptide, we used pharmacological inhibitors of NOX isoforms. First, the non-selective NOX inhibitor, apocynin, and the NOX1/4-selective inhibitors, setanaxib and GKT136901, did not affect the baseline levels of hydrogen peroxide (Figure 4A–C). Similarly to the previous result (Figure 1B), treatment of endothelial cells with 2 µM Aβ_40_ significantly increased hydrogen peroxide production (Figure 4A–C). This Aβ_40_-induced hydrogen peroxide production was inhibited concentration-dependently by the non-selective NOX inhibitor, apocynin (Figure 4A), and NOX1/4-selective inhibitors, setanaxib (Figure 4B) and GKT136901 (Figure 4C). To test the possibility that these NOX inhibitors decrease the hydrogen peroxide formation via their direct antioxidative activity, the DPPH free-radical scavenging assay was performed. The naturally occurring polyphenol, quercetin, showed its potent free-radical scavenging activity with a calculated EC_50_ value of 10.3 ± 0.8 µM (Figure 4D). However, apocynin had no free-radical scavenging activity on DPPH free radicals (EC_50_: >200 µM), whereas setanaxib and GKT136901 revealed weak free-radical scavenging activities with calculated EC_50_ values of 51.4 ± 7.4 µM and 30.1 ± 2.7 µM, respectively.

### 2.5. NOX4 Inhibition Did Not Affect the Intracellular Levels of Superoxide in bEnd.3 Cells

Next, we examined the role of NOX4 in the production of intracellular superoxide utilizing the superoxide-sensitive dye, MitoTracker Red CM-H2XRos. The fluorescent signals were present within the mitochondria of endothelial cells with or without treatment with 2 µM Aβ_40_ (Figure 5A, panels b and c). To further quantify the superoxide-sensitive fluorescent signals, the averaged fluorescent intensity within the captured images (Figure 5A) was normalized with the number of cells within the same field (Figure 5B–D). We found that treatment of endothelial cells with 2 μM Aβ_40_ for 3 h increased the MitoTracker fluorescent intensity as compared with the control group (L-15 medium only). (Figure 5B–D) We observed that the pharmacologic inhibitors of NOX, apocynin (Figure 5B), setanaxib (Figure 5C), and GKT136901 (Figure 5D) did not affect the superoxide-sensitive signals in the presence and absence of Aβ_40_ (*p* > 0.05 vs. the control group).

## 3. Discussion

The present study demonstrated that Aβ_40_ peptide exerts beneficial effects in bEnd.3 mouse brain endothelial cells via the NOX4-dependent mechanism. We found that Aβ_40_ enhances the release of ROS (H_2_O_2_ and O_2_^•−^ to a lesser degree) and NO, and cell viability. Our data revealed that bEnd.3 cells predominantly express *Nox4* mRNA among NOX1-4 isoforms. The increased release of H_2_O_2_ induced by Aβ_40_ was attenuated in the presence of the pan-NOX inhibitor, apocynin, or the NOX1/4-selective inhibitors, setanaxib and GKT136901. Taken together, these data indicate that NOX4-derived H_2_O_2_ contributes to the protective actions of Aβ_40_ in these cultured cells.

We and others previously found that NOX2-dependent ROS causes cerebrovascular dysfunction and progression of CAA in the mouse model of AD [6,33]. Since other NOX isoforms, such as NOX1, NOX3, and NOX4, are also expressed in the brain, and their selective inhibitors have been discovered recently, we sought to examine the role of each NOX isoform in cerebral endothelial function in this study. The results from RT-qPCR indicate that the bEnd.3 endothelial cells express exclusively the NOX4 isoform (Figure 3). Since multiple NOX isoforms are known to be expressed in the cerebrovasculature [6,26], these unexpected results allowed us to explore the physiological and/or pathophysiological role of NOX4 isoform-specific ROS production using this cultured cell model. We measured the release of H_2_O_2_ for 3 h of incubation with Aβ_40_-peptide since we previously observed that monomeric Aβ_40_ is stable and remains as soluble monomers in the L-15 incubation medium for this period [32,35], unlike Aβ_42_ which is prone to rapidly aggregate within 3 h [36]. The increase in cell viability observed following Aβ_40_ treatment is likely attributable to enhanced metabolic activity rather than proliferation. This interpretation is supported by the use of a resazurin-based assay, which detects redox-driven metabolic activity, and by the serum-free conditions used, which promote growth arrest in bEnd.3 cells. These findings suggest that NOX4-derived H_2_O_2_ may enhance cell survival via redox-sensitive adaptive signaling. Future studies using proliferation-specific assays will be necessary to confirm this distinction. When bEnd.3 cells were directly treated with H_2_O_2_, cell viability was also significantly increased in the presence of low concentrations (10–100 µM) of H_2_O_2_, whereas bEnd.3 cells were not tolerant to 10 mM H_2_O_2_ (Figure 2). Consistently, we found that both Aβ_40_ and H_2_O_2_ enhanced NO bioavailability, suggesting H_2_O_2_ as a potential downstream mediator of the vasoprotective action of Aβ_40_. Though a variety of Aβ species, including monomers, oligomers, and fibrils, are associated with AD pathogenesis and cerebrovascular dysfunction, our results are in line with previous reports showing that monomeric Aβ has physiological roles in maintaining the integrity and functions of the cerebrovasculature [37,38]. In particular, soluble Aβ_40_ is a vasoactive peptide that controls cerebrovascular tone [11].

To our knowledge, our results are the first report that NOX4-driven H_2_O_2_ is involved in mediating the beneficial effect of Aβ_40_ in cerebrovascular endothelial cells. In contrast to these findings, we have previously discovered that NOX2-driven superoxide radical is the key contributor to cerebrovascular dysfunction and CAA deposition in the Tg2576 transgenic mouse model [6], suggesting distinct roles of NOX isoforms in normal physiology as well as AD pathogenesis in the brain. Superoxide immediately reacts with the potent vasodilator NO, reducing its bioavailability in the vasculature. In addition, the product of the reaction between superoxide and NO is peroxynitrite (ONOO-), which is another species of potent free radical that can oxidize many cellular components, including proteins, lipids, and nucleic acids [21,22,29]. In contrast to NOX2 (and other isoforms) that produce superoxide, NOX4 predominantly generates hydrogen peroxide [29]. In the present study, we observed that the non-selective NOX inhibitor apocynin and the NOX1/4-selective inhibitors setanaxib and GKT136901 inhibited Aβ_40_-induced hydrogen peroxide production in a concentration-dependent fashion. Since bEnd.3 endothelial cells express only the NOX4 isoform (Figure 3), our data indicate that NOX4 is responsible for the generation of hydrogen peroxide in response to Aβ_40_. It has been reported that small-molecule NOX inhibitors have antioxidant properties by inhibiting the catalytic activity of NOX enzymes and/or directly scavenging free radicals. To test the latter possibility, we performed the DPPH free-radical scavenging assay in a cell-free condition. We found that, unlike the polyphenolic antioxidant quercetin, the pan-NOX inhibitor apocynin did not possess antioxidant capacity (Figure 4D). Likewise, both setanaxib and GKT136901 had little effect on scavenging DPPH free radicals. Taken together, these data indicate that these NOX inhibitors have antioxidant properties by directly inhibiting the catalytic activity of NOX enzymes. We observed that the superoxide levels were also increased in the presence of Aβ_40_; however, none of the NOX inhibitors were able to attenuate the superoxide production. Since other redox systems are likely present in endothelial cells, further studies are necessary to identify the mechanism underlying NOX-independent superoxide production. Nevertheless, our data suggest that NOX4 is responsible for the production of hydrogen peroxide in response to Aβ_40_.

Growing lines of evidence suggest that NOX4-driven H_2_O_2_ plays protective roles in many organs and tissues. For example, NOX4 deficiency is associated with hypertension and potentiation of endothelial dysfunction and vascular remodeling in a mouse model of hypertension [39]. Recent in vitro and in vivo studies demonstrate that NOX4 plays a key role in endothelial proliferation, adhesion, and revascularization following ischemia [38,39]. Nox4 deficiency attenuated ischemia-induced angiogenesis and potentiated responses to angiotensin II. The vasoprotective effects of NOX4 are thought to be mediated by H_2_O_2_ generation, maintenance of NO production, and increased expression of the antioxidant heme oxygenase-1 [31]. Another study demonstrates that NOX4 has neuroprotective effects by regulating ROS and calcium homeostasis and thereby preventing hyperexcitability and, consequently, neuronal death in cultured neurons [40]. In contrast, dysregulation of the NOX4 activity plays a detrimental role in various pathological conditions, including metabolic diseases, neurodegenerative diseases, and cancer [29,34]. Taken together, the role of NOX4 may be cell-type-specific, complex, and dependent on the experimental settings; therefore, the therapeutic potential of the anti-ROS approach targeting NOX4 should be carefully assessed under normal physiological and pathophysiological conditions in different organs and tissues.

The mechanism by which Aβ_40_ enhances the release of hydrogen peroxide in an NOX4-dependent fashion remains to be determined. NOX1-3 isoforms require interaction with p22^phox^ transmembrane protein along with the other cytosolic subunits for their catalytic activity to generate primarily superoxide radical [29]. However, NOX4 requires only the p22^phox^ subunit to be constitutively active to generate hydrogen peroxide [29,30,31]. Since the treatment of bEnd.3 cells with Aβ_40_ did not affect the levels of NOX4 mRNA (Figure 3B), the increase in the NOX4 expression is not likely responsible for the increased NOX4 activity induced by Aβ_40_. Interestingly, Alves-Lopes and colleagues reported that angiotensin II immediately increased NOX4-dependent hydrogen peroxide production in rat aortic endothelial cells [39], suggesting a possible signaling cascade that immediately leads to NOX4 activation upon stimulation of angiotensin receptors. Similarly, our findings suggest that Aβ_40_ increases the NOX4 catalytic activity through an unknown mechanism. It may be possible that an interaction of Aβ_40_ with a specific type of cell-surface receptor is involved in Aβ_40_-induced NOX4 activation. We and others found that different Aβ_40_ species can interact with a variety of cell-surface receptors, including heparan sulfate proteoglycans, scavenger receptors, and low-density lipoprotein receptor-related protein 1 (LRP1) [6,32,35,41]. We, therefore, plan to identify which cell-surface receptors for Aβ are responsible for NOX4 activation. In addition, it would be interesting to further explore the downstream mediators of NOX4-driven H_2_O_2_ in response to Aβ_40_. We previously reported that Aβ_40_ increases the Ca^2+^ influx in cerebrovascular smooth muscle [32]. Alves-Lopes et al. recently reported that NOX4-derived H_2_O_2_ increases the Ca^2+^ influx through a non-selective cation channel and the release of NO by eNOS in rat aortic endothelial cells [39]. Similarly, a previous report demonstrates that Aβ_40_ stimulates the NO pathway in primary cultured endothelial cells from bovine aorta and rat coronary microvessel [42]. We, therefore, speculate that monomeric Aβ_40_ plays a role in the normal physiology of cerebrovascular endothelial cells through NOX4-dependent H_2_O_2_ release, and the regulation of intracellular Ca^2+^ levels and eNOS activity. Further studies are necessary to test this possibility.

While this study establishes a mechanistic link between Aβ_40_ and NOX4-derived H_2_O_2_ in murine brain endothelial cells, the relevance of this pathway in the human AD brain remains to be clarified. Recent advances in transcriptomic profiling of the AD brain, particularly through single-cell and spatial RNA sequencing approaches, have begun to unravel cell-type-specific gene expression signatures in both vascular and nonvascular compartments. For example, a recent systematic review of single-cell studies in dementia highlights that endothelial and other vascular-associated cell types show transcriptional alterations in AD and other dementias, supporting the feasibility of examining NOX4 expression in clinical vascular datasets [43]. Additionally, NOX4 has been implicated in pathologic astrocyte ferroptosis in AD, indicating that its role may be cell-type-specific and context-dependent [43,44]. The potential for NOX4 to exert both protective and detrimental effects depending on cell type and disease stage underscores the importance of integrating future in vitro findings with transcriptomic data from human brain tissue. Indeed, single-cell and spatial transcriptomic studies are rapidly advancing our understanding of AD gene networks and could be used to determine whether NOX4 is enriched or dysregulated in cerebrovascular endothelial populations and whether it co-expresses with Aβ-binding receptors [45]. These approaches will be important for defining the translational relevance of NOX4-dependent redox signaling in human AD pathology.

One of the limitations of this study was the use of the bEnd-3 endothelial cell line since its biological response to Aβ_40_ may differ from that of primary cultured endothelial cells. Therefore, the results found in this study would still need to be further investigated in a primary endothelial culture system and/or an in vivo animal model to understand the exact role of each NOX isoform in Aβ_40_ pathophysiology. Nevertheless, our findings indicate that bEnd-3 cells would provide a unique endothelial cell model to study the role of NOX4 since other NOX isoforms are not expressed in this cell line. As stated above, the next steps could be to further investigate these findings in in vivo studies. It would be interesting to further investigate whether NOX4 plays a normal physiological and/or pathological role in the cerebrovascular pathophysiology associated with AD and the therapeutic effects of selective NOX4 treatment of AD.

## 4. Materials and Methods

### 4.1. Materials

The mouse brain endothelial cell line, bEnd.3 cells, was purchased from American Type Culture Collection (ATCC, Rockville, MD, USA). Amplex red, horseradish peroxidase, apocynin and catalase were obtained from MiliporeSigma (Burlington, MA, USA). Setanaxib and GKT136901 were obtained from MedChemExpress (Monmouth Junction, NJ, USA). Recombinant human Aβ_40_ peptide purified with 1,1,1,3,3,3-hexafluoro-2-propanol (HFIP) was purchased from MiliporeSigma. Dulbecco’s modified eagle medium (DMEM), Leibovitz’s L-15 medium, fetal bovine serum (FBS), 0.25% trypsin-EDTA solution, and cell culture plates were purchased from Thermo Fisher Scientific (Waltham, MA, USA).

### 4.2. Cell Culture

The mouse brain microvessel endothelial cells, bEnd.3 cells, were used in the study. Endothelial cells were cultured in a cell culture flask in DMEM supplemented with 10% FBS, 100 U/mL penicillin and 100 μg/mL streptomycin (MiliporeSigma; Burlington, MA, USA) at 37 °C in a humidified 5% CO_2_ incubator (Thermo Fisher Scientific; Waltham, MA, USA). Cell cultures were passaged no more than six times during these experiments to ensure consistency [36].

### 4.3. Detection of Hydrogen Peroxide

The amplex red hydrogen peroxide assay was performed per the previous protocol [32] with modifications. Cells were grown in a 96-well black clear-bottom plate until they reached near 100% confluence. Each well was washed once with 200 μL warm L-15 medium. The master reaction buffer was prepared in L-15 medium containing 0.05% bovine serum albumin, 10 µM amplex red, and 1 U/mL horseradish peroxidase. After washing with warm L-15 medium, bEnd.3 cells were incubated at 37 °C in the 100 µL reaction buffer in the presence of Aβ_40_ and other reagents indicated. Three hours later, the fluorescent intensity was measured at an excitation wavelength of 530 nm and an emission wavelength of 590 nm at room temperature using a Biotek Synergy HT plate reader (Winooski, VT, USA).

### 4.4. Detection of Nitric Oxide

Nitric oxide (NO) measurement was conducted per the protocol previously established [37,46]. Cells were grown in a 96-well black clear-bottom plate until they reached near 100% confluence. After washing with warm L-15 medium, cells were incubated with 100 µL of L-15 medium containing the NO-sensitive dye, 4-amino-5-methylamino-2′,7′-difluorofluorescein diacetate (DAF-FM; Thermo Fisher Scientific; Waltham, MA, USA) in a final concentration of 5 µM. Thirty minutes later, cells were rinsed with warm L-15 medium twice, treated with the reaction buffer containing L-15 medium, 0.05% BSA, Aβ_40_ and other reagents indicated. The DAF fluorescent intensity was measured at an excitation wavelength of 485 nm and an emission wavelength of 520 nm.

### 4.5. Reverse Transcription-Quantitative Polymerase Chain Reaction (RT-qPCR)

To quantify the expression levels of gene transcripts, we performed an RT-qPCR assay per our published protocol [32,47]. Endothelial bEnd.3 cells were grown in a 6-well plate and treated with Aβ_40_ and other reagents for various time points indicated. Cells were rinsed with ice-cold phosphate-buffered saline (PBS) twice and 1 mL/well of TRIzol (Thermo Fisher Scientific (Waltham, MA, USA) was added before cells were stored at −80 °C. Samples were thawed and incubated at room temperature for 5 min. After the TRIzol solution was transferred from the plate to a microcentrifuge tube, 200 μL 1-bromo-3-chloropropane was added to each tube. Samples were spun down in a centrifuge at 12,000× *g* for 15 min at 4 °C. The top clear aqueous layer was transferred to a new tube, and 500 μL 100% isopropanol was added to each tube, followed by adding 1 μL Glyco Blue solution. Samples were then spun down with a bench-top centrifuge at 12,000× *g* for 10 min at 4 °C. Supernatant was removed, leaving the RNA pellet. Then 1 mL of 80% ethanol was added to each tube and then spun at 12,000× *g* for 10 min at 4 °C. The RNA pellet was resuspended in 30 μL diethyl pyrocarbonate (DEPC)-treated water and triturated until the pellet was no longer visible. The RNA content of each sample was measured using a Nanodrop One spectrophotometer (Thermo Fisher Scientific). One microgram of RNA was used to make cDNA using an iScript cDNA synthesis kit (Bio-Rad; Hercules, CA, USA). Quantitative PCR was performed using an SYBR green supermix kit according to the manufacturer’s protocol (Bio-Rad; Hercules, CA, USA) with a CFX Connect real-time PCR system. The following PCR primers were used: *Gapdh* (forward: 5′-CTTTGTCAAGCTCATTTCCTGG-3′; reverse: 5′-TCTTGCTCAGTGTCCTTGC-3″), *Nox1* (forward: 5′-CTTTGTCAAGCTCATTTCCTGG-3′; reverse: 5′-TCTTGCTCAGTGTCCTTGC-3′), *Nox2* (forward: 5-TCCTATGTTCCTGTACCTTTGTG-3′; reverse: 5′-GTCCCACCTCCATCTTGAATC-3′), *Nox3* (forward: 5′-TGCCCTGTACCTCAATTTTCTG-3′; reverse: 5′-ACACGCATACAAGACCACAG-3′), and *Nox4* (forward: 5′-TCCAAGCTCATTTCCCACAG-3′; reverse: 5′-CGGAGTTCCATTACATCAGAGG-3′). The levels of mRNA expression relative to *Gapdh* mRNA were calculated using the comparative *C_t_* method.

### 4.6. Cell Viability Assay

The bEnd.3 cell viability was assessed by the resazurin-based fluorescent method [48]. Cells were grown in a 96-well black clear-bottom plate until they were near confluent, between 60 and 70%. Cells were rinsed once with 200 μL warm L-15 medium, and 100 μL of serum-free L-15 medium containing various reagents was added. The plate was incubated at 37 °C in a humidified 5% CO_2_ incubator overnight (16–18 h). On the next day, the L-15 medium containing 0.3 mg/mL resazurin was added to each well, followed by incubation for 3 h. The fluorescent intensity of the resazurin metabolite, resorufin, was measured at an excitation wavelength of 530 nm and an emission wavelength of 590 nm at room temperature.

### 4.7. Detection of Superoxide

The superoxide measurement assay was performed per the previous protocol [11,32] with modifications. Cells were grown in a 96-well black clear-bottom plate until they reached approximately 100% confluence. Cells were incubated with the ROS-sensitive dye, MitoTracker Red CM-H2XRos (Thermo Fisher Scientific; Waltham, MA, USA) for 30 min. Cells were rinsed with warm L-15 medium, followed by treatment with Aβ_40_ and other reagents. Three hours later, the ROS signals were photographically captured and quantified using a BioTek Cytation5 fluorescent imaging system (Winooski, VT, USA).

### 4.8. 2,2-Diphenyl-1-picrylhydrazyl (DPPH) Assay

Free-radical scavenging activity was determined per our published protocol [49]. In a 96-well plate, 50 µL of 0.2 mM 2,2-diphenyl-1-picrylhydrazyl (Sigma-Aldrich; Burlington, MA, USA) dissolved in ethanol was mixed with an equal volume of 0.1 M Tris-HCl buffer (pH 7.4) containing various concentrations of test compounds. After incubation for 20 min, O.D. was read at 517 nm.

### 4.9. Data Analysis

One-way or two-way analysis of variance (ANOVA) was used to compare data across experimental groups using the GraphPad Prism Version 10 software (Boston, MA, USA). Statistical significance was defined as *p* < 0.05.

## Figures and Tables

**Figure 1 ijms-26-06759-f001:**
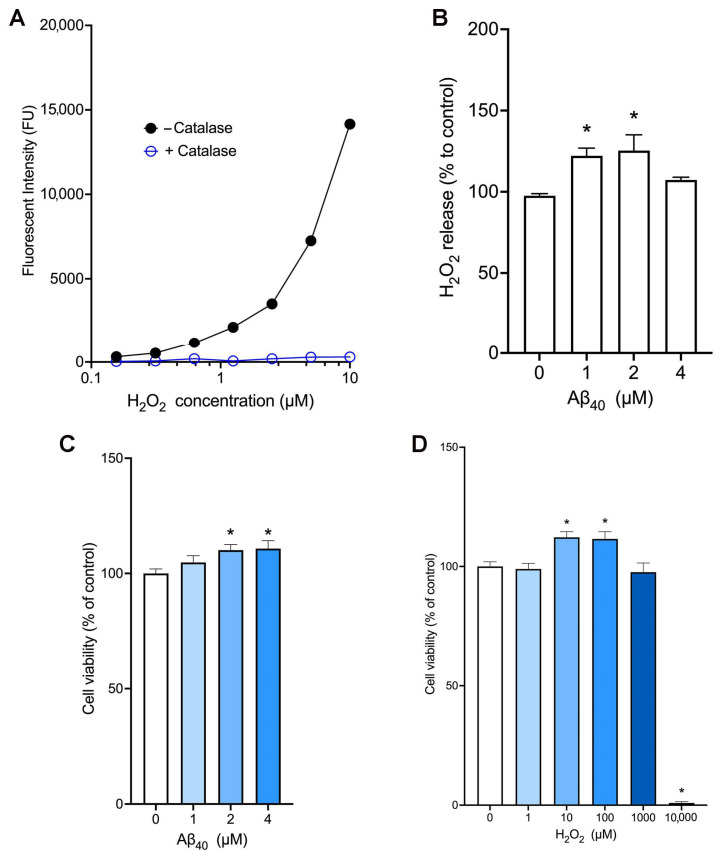
Aβ_40_ increases the release of hydrogen peroxide and cell viability in bEnd.3 cells. (**A**) The amplex red assay was performed in a cell-free condition with various concentrations of hydrogen peroxide in the presence or absence of catalase (*n* = 2). (**B**,**C**) Endothelial cells were incubated with the L-15 culture medium containing Aβ_40_ at 37 °C. Hydrogen peroxide release (**B**) and cell viability (**C**) were determined 3 h and 24 h after the treatment, respectively. (**D**) Endothelial cells were incubated with various concentrations of hydrogen peroxide for 24 h, and cell viability was measured. Data indicate mean ± S.E.M. from three independent experiments with quadruplicate samples. *: *p* < 0.05 as compared with the control group determined by one-way ANOVA followed by a post hoc analysis.

**Figure 2 ijms-26-06759-f002:**
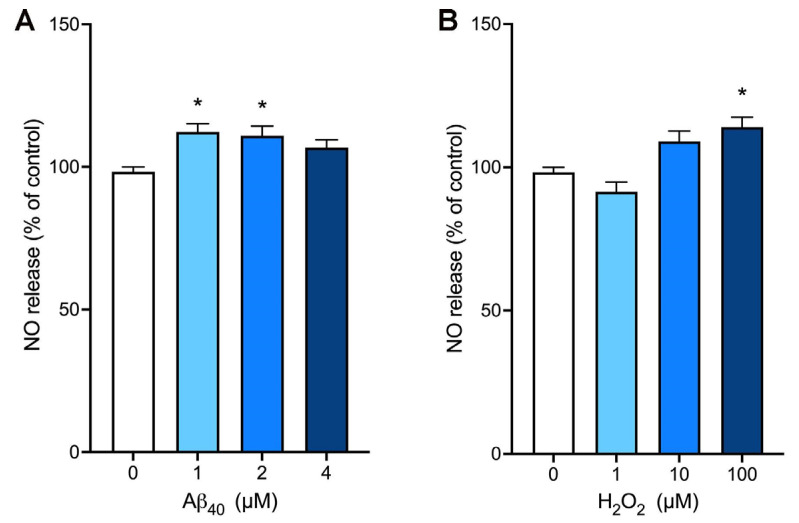
The production of nitric oxide is enhanced by Aβ_40_ and hydrogen peroxide in bEnd.3 cells. Endothelial cells were preloaded with the nitric oxide (NO)-sensitive dye, DAF-AM, and incubated with various concentrations of Aβ_40_ (**A**) or hydrogen peroxide (**B**) for 3 h. Data indicate mean ± S.E.M. from three independent experiments with quadruplicate samples. *: *p* < 0.05 as compared with the control group determined by one-way ANOVA followed by a post hoc analysis.

**Figure 3 ijms-26-06759-f003:**
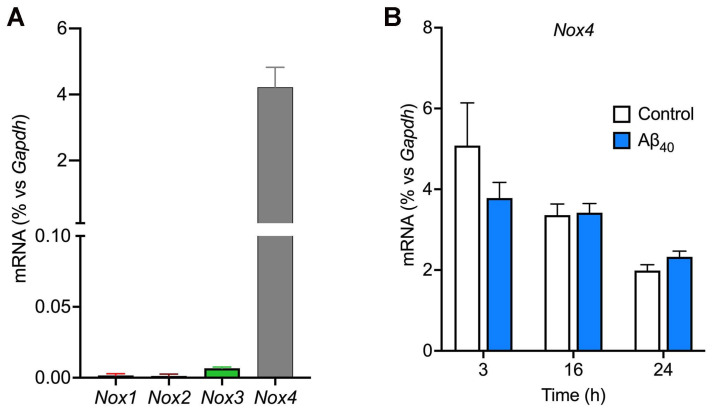
Evidence that bEnd.3 endothelial cells predominantly express the *Nox4* gene. (**A**) The levels of gene expression of *Nox* isoforms were determined by the real-time quantitative polymerase chain reaction method. (**B**) Endothelial cells were incubated with or without 2 µM Aβ_40_ for 3, 16, and 24 h, and expression levels of *Nox4* mRNA were quantified. Data indicate mean ± S.E.M. (*n* = 4).

**Figure 4 ijms-26-06759-f004:**
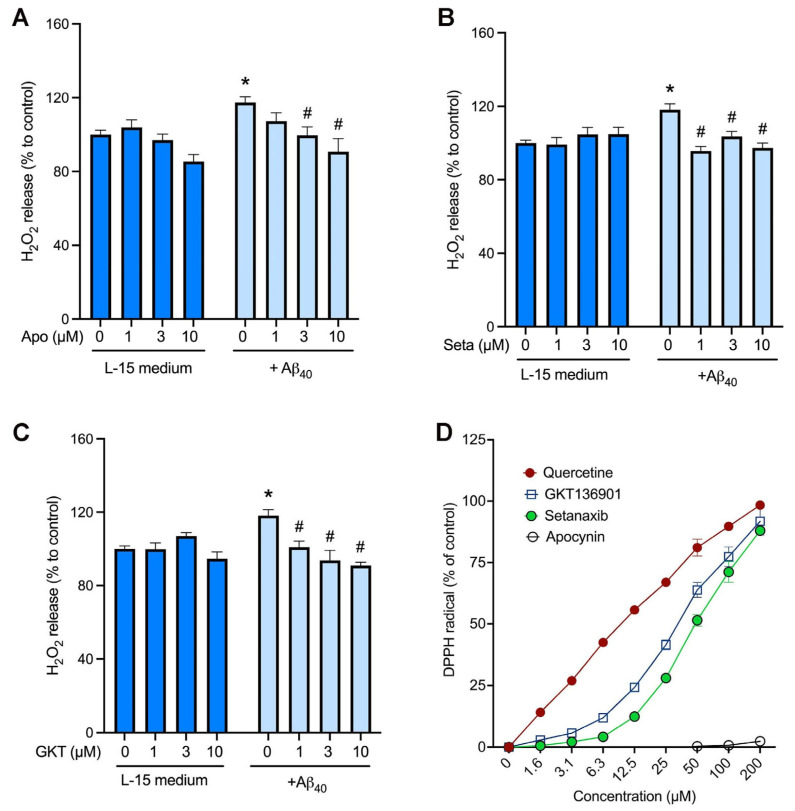
Inhibition of NOX4 significantly attenuates Aβ_40_-induced hydrogen peroxide production in bEnd.3 cells. (**A**–**C**) Cells were incubated with the hydrogen peroxide assay buffer (L-15 medium) or 2 μM Aβ_40_ in the presence of various concentrations of the non-selective NOX inhibitor, apocynin (**A**), and the NOX4-selective inhibitors, setanaxib (Seta; (**B**)) and GKT136901 (GKT; (**C**)). Three hours later, hydrogen peroxide production was measured by the amplex red method. Data indicate mean ± SEM (*n* = 12). * *p* < 0.05 vs. control group. # *p* < 0.05 vs. Aβ_40_ alone group. Analyzed by two-way ANOVA followed by a multiple comparison test. (**D**) The antioxidant activity of NOX inhibitors was assessed by the DPPH free-radical assay. The polyphenolic antioxidant, quercetin, was used as a positive control.

**Figure 5 ijms-26-06759-f005:**
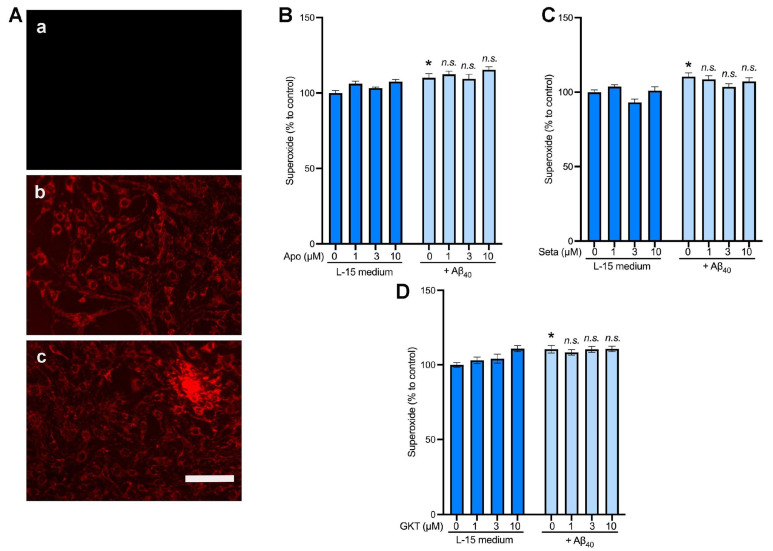
The effects of NOX4 inhibitors on superoxide production induced by Aβ_40_ in bEnd.3 cells. Cells were preloaded with the superoxide-sensitive dye, MitoTracker Red, for 30 min. Cells were then incubated for 3 h with the L-15 culture medium with or without 2 μM Aβ_40,_ and the fluorescent images of the MitoTracker Red were photographically captured. (**A**) Representative images of cells without the superoxide dye (**a**); the superoxide dye alone (**b**); and the superoxide dye plus 2 μM Aβ_40_ (**c**). (**B**–**D**) Cells were incubated with the assay buffer (L-15 medium) or 2 μM Aβ_40_ in the presence of various concentrations of the non-selective NOX inhibitor, apocynin (**B**), and the NOX4-selective inhibitors, setanaxib (Seta; (**C**)) and GKT136901 (GKT; (**D**) Three hours later, the superoxide levels were quantified. Data indicate mean ± S.E.M. (*n* = 12). * *p* < 0.05 vs. control group. n.s.: not significant. Analyzed by two-way ANOVA followed by a multiple comparison test.

## Data Availability

All data are available on reasonable request from the corresponding author.

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
