# Peer review of "40 Improves Cerebrovascular Endothelial Function via NOX4-Dependent Hydrogen Peroxide Release"

_ijms, 2025, doi:10.3390/ijms26146759_

Round 1

Reviewer 1 Report

Comments and Suggestions for Authors

This is a well-designed, well-documented, and generally well-presented study on the fole of NOX4 in Ab40-stimulated protection of bEnd.3 endothelial cells. The findings are clear and significant and the limitations of the study are appropriately discussed.  My only concern is that the Discussion section was made unnecessarily long and burdensome by exhaustive reiteration of the results.  The actual discussion of the results in the discussion section is good and should be kept, just not the rehasing of what is already clear in the Results section.

Reviewer 2 Report

Comments and Suggestions for Authors
  1. The study investigates the effect of Aβ40 on cell viability, which might be influenced by cell proliferation. Is there evidence to differentiate whether the observed changes in cell viability are due to enhanced cell metabolism or an actual increase in cell number?
  2. Figure 3 demonstrates that NOX4 is the predominant NOX isoform in bEnd.3 cells. This prompts a question about whether NOX4 expression is associated with cell adhesion and growth in vitro.
  3. The manuscript does not incorporate relevant clinical single-cell sequencing data. Would it be feasible to explore if the gene expression patterns and interactions found in this study are supported by clinical data, potentially enriching the understanding of the gene interaction network in Alzheimer's disease?
  4. The interaction between Aβ40 and NOX4 is not fully clarified. While the study shows that NOX4 mediates the Aβ40-induced increase in Hâ‚‚Oâ‚‚ production, is there a direct interaction between Aβ40 and NOX4, or does Aβ40 activate NOX4 via alternative signaling pathways or molecules?

Round 2

Reviewer 2 Report

Comments and Suggestions for Authors

I believe that the revisions have strengthened the manuscript and addressed the key issues raised during the review process. I am pleased to inform you that I am considering the acceptance of your manuscript.